# Conceptualize Any Network:
# A Concept Extraction Framework for
# Holistic Interpretability of Image Classifiers

## Abstract

Attribution-based and concept-based methods dominate the area of post-hoc explainability for vision classifiers. While attribution-based methods highlight crucial regions of the input images to justify model predictions, concept-based methods provide explanations rooted in high-level properties that are generally more understandable for humans. In this work, we introduce "Conceptualize Any Network" (CAN), a comprehensive post-hoc explanation framework that combines the wide scope of feature attribution methods and the understandability of concept-based methods. Designed to be model agnostic, CAN is capable of explaining any network that allows for the extraction of feature attribution maps, expanding its applicability to both CNNs and Vision Transformers (ViTs). Moreover, unlike existing concept-based methods for vision classifiers, CAN extracts a set of concepts shared across all classes, enabling a unified explanation of the model as a whole. Extensive numerical experiments across different architectures, datasets, and feature attribution methods showcase the capabilities of CAN in Conceptualizing Any Network faithfully, concisely, and consistently. Furthermore, we manage to scale our framework to all of ImageNet's classes which has not been achieved before.

## 1 Introduction

The recent developments in Computer Vision improve the prediction accuracy of the new models for increasingly sophisticated tasks. However, it comes with the cost of less transparent architectures that are not fully understandable even by experts, leading to concerns about the usage of such models (Zhao et al., 2023; Hamon et al., 2020) which necessitate the introduction of new legislation (Kaminski & Urban, 2021; Veale & Zuiderveen Borgesius, 2021).

To alleviate these concerns, researchers have developed different methods to explain decisions of given pretrained models, collectively termed as *post-hoc explainability methods*. The most prominent of these are feature attribution approaches (Selvaraju et al., 2017; Sundararajan et al., 2017; Shrikumar et al., 2017; Ribeiro et al., 2016; Binder et al., 2016) that output an importance map over the space of input features, i.e. pixels. However, the use of pixel space for generating visual saliency maps has been criticized for multiple reasons. Specifically, saliency methods are criticized for only highlighting "where" a model focuses and poor at identifying "what" features the model extracts i.e. the underlying semantic patterns (Colin et al., 2022). Moreover, they locally interpret a model and are incapable of deriving a global understanding of the decision-making process.

These limitations have led to the growing prominence of concept-based interpretability approaches that aim to extract and interpret the given model via a dictionary of high-level concept representations (Kim et al., 2018; Ghorbani et al., 2019; Vielhaben et al., 2023). While a variety of methods have been proposed to this end, they explicitly make assumptions about the model architecture (Fel et al., 2023; Vielhaben et al., 2023) and rely on both, selecting and accessing, the right internal representations of the given model (Ghorbani et al., 2019). Thus, the prior methods generalize poorly in terms of the architectures they apply to, as evidenced by the vast majority of prior approaches only being applied to CNNs.

In this paper, we propose a novel generic framework to "Conceptualize Any Network" (CAN). Our framework makes no prior assumption about the model architecture and can be generalized to any network. It relies on a feature attribution algorithm to extract information relevant to the given model. This information is clustered in the activation space of a fixed encoder to discover the concept dictionary. Unlike many of the previous approaches (Kim et al., 2018; Ghorbani et al., 2019; Fel et al., 2023) that extract concept dictionaries for each class separately, our method extracts a shared concept dictionary across all classes, thus providing a holistic understanding of the model's decisions. Our key contributions can be summarized as:

- We present a novel post-hoc interpretability method able to extract a dictionary of concepts from any feature attribution map defined from a pretrained model. This versatility allows to cope with arbitrary model architectures (CNNs, ViTs, etc.).

- Our method provides a holistic view of the model by extracting a single concept dictionary shared between all classes. We demonstrate this capability at scale by extracting dictionaries for all ImageNet classes. To the best of our knowledge, among the similar methods, ours is the first approach applied to this scale.

- We extensively validate our approach quantitatively and demonstrate that it extracts faithful, concise and relevant explanations. Our experiment to demonstrate the relevance of the concept dictionaries to model's output also provides a principled way of selecting the concept dictionary size.

## 2 RELATED WORK

Besides post-hoc interpretation, feature attribution is also commonly used as a means of interpretation for networks interpretable by design, such as for Contextual Explanation Networks (Al-Shedivat et al., 2020) and B-cos Networks (Böhle et al., 2024). In contrast to feature attribution approaches, our method is a concept based interpretability approach that uses outputs of an underlying feature attribution method to build its concept dictionary.

**Concept activation vector (CAV) approaches** Kim et al. (2018) first proposed the notion of concept activation vectors (CAVs) to represent concepts in the activation space of a deep neural network classifier. The concepts are defined as a set of user-provided examples. They propose to represent the concept in the activation space of a neural network by finding a hyperplane in a given layer that separates the specified set of examples from a random set, defined as the CAV. Ghorbani et al. (2019) proposed ACE, that further built on this approach by automating the concept extraction process. They build a concept dictionary for a given class by extracting superpixels at various resolutions for a given set of samples (from the class), and clustering them in the activation space. The centroids of the clusters represent the different CAVs. Fel et al. (2023) instead proposed to decompose activations from image crops of a class with NMF, to learn a dictionary of CAVs, termed as CRAFT. Concept-SHAP (Yeh et al., 2020) introduced the notion of completeness score that estimates the extent to which extracted concepts explain the prediction of the classifier. MCD (Vielhaben et al., 2023) also introduces another version of the completeness score based entirely on the model parameters. A unifying framework for concept extraction covering most of the prior CAV-based approaches was presented by Fel et al. (2024), which essentially considers all the approaches as instantiation of a dictionary learning problem.

All the prior approaches however make assumptions about the underlying architecture of the classifier and inherently rely on the internal representations. Moreover, in practice, almost all are only applied on convolutional neural networks (CNNs), with the exception of MCD which can also be applied for certain vision transformers not using a CLS token (Vielhaben et al., 2023). In contrast, we make no assumption about the internal architecture of the visual classifier. We assume that we have access to the output of a feature attribution method. If the classifier is differentiable, various candidates for such feature attribution methods exist. Even if not, a black-box feature attribution method could be used. Our method can be particularly useful to understand proprietary models via concept based explanations if a feature attribution output is accessible via an API. We also design our method to extract a shared concept dictionary for *all* classes simultaneously, an aspect that has been experimentally explored very briefly in prior CAV-based approaches. These differences with prior works are summarized in Table 1.

|  | ACE | CRAFT | ConceptSHAP | MCD | CAN *(Ours)* |
|---|---|---|---|---|---|
| Multi-class | ✗ | ✗ | ✓ | ✓ | ✓ |
| Multi-architecture | ✗ | ✗ | ✗ | - | ✓ |

Table 1: Summary of concept discovery methods in the litterature with their limitations.

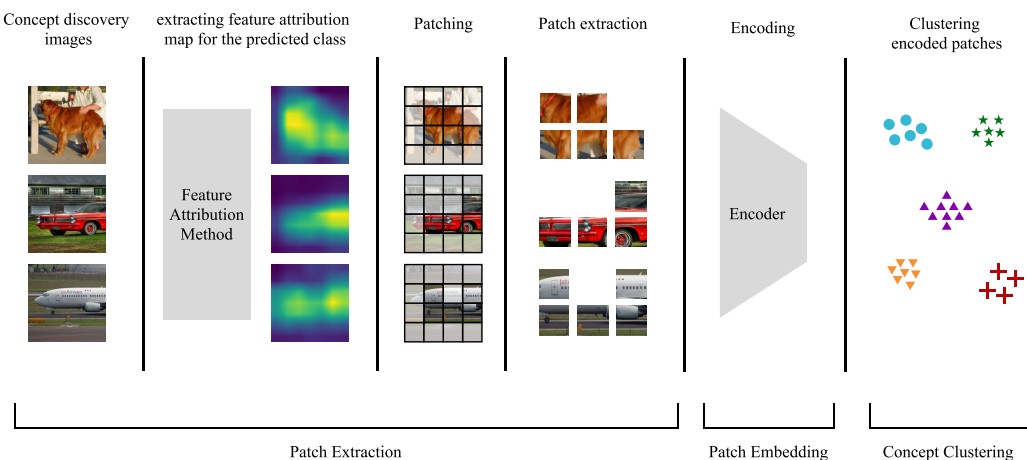

Figure 1: A high-level overview of *concept discovery* in CAN framework. A set of discovery images and their feature attribution maps are divided into patches. Guided by the feature attribution maps, the most important patches, whose accumulated weighted sum is below a threshold, are extracted and passed through an encoder. The embedded patches are then clustered to extract concepts.

## 3 METHODS

This section presents our concept-based framework CAN. We are interested in explaining a given pre-trained classification model $f : \mathbb{R}^{d_1} \to \{1, \dots, l\}$. In our framework, the *explainability* task is divided into two steps. At **discovery** time, the *concept discovery* algorithm, $\mathcal{D}$, extracts a set of concepts from the predictive model at hand, $f$, and a "training" dataset $\mathbb{X}_{\text{disc}}$ dedicated to the discovery task. At **testing** time, the *concept assignment* algorithm, $\mathcal{A}$, leverages the set of discovered concepts to identify for each input image of a "test" dataset $\mathbb{X}_{\text{test}}$ the concepts important to the prediction provided by $f$.

In the next two sections, we describe each algorithm and their components.

### 3.1 CONCEPT DISCOVERY

Define the *discovery* dataset as $\mathbb{X}_{\text{disc}} = \{(\mathbf{x}_i, y_i) \in \mathbb{R}^{d_1} \times \{1, \dots, l\}\}_{i=1}^{N_{\text{disc}}}$ of $N_{\text{disc}}$ images along with their class labels. The goal of *concept discovery* $\mathcal{D}$ is to extract a joint set of $k$ concepts $\mathbb{D}_{\text{disc}} = \{\mathbf{c}_n \in \mathbb{R}^{d_2}\}_{n=1}^{k}$, that will constitute our *concept dictionary* to explain $f$, and a *class-concept importance matrix* $\mathbf{W}^{\text{disc}} \in [0, 1]^{l \times k}$ from the dataset $\mathbb{X}_{\text{disc}}$.
Note that we are interested in extracting relevant concepts for $f$ that are *shared between all classes*, instead of extracting concepts for *each class separately*. In this work, we define concepts $\mathbf{c}_n$ as *centroids of clusters*, computed from relevant parts of inputs, here patches cropped from the original images, embedded in a lower-dimensional representation space $\mathbb{R}^{d_2}, d_2 < d_1$.

As illustrated in Figure 1, we further define the *concept discovery* function $\mathcal{D}$ as the composition of three functions, *patch extraction* $\mathcal{E}_1$, *patch embedding* $\mathcal{E}_2$ and *concept clustering* $\mathcal{D}_3$, that each have their own hyperparameters. We describe each of these functions below.

### 3.1.1 PATCH EXTRACTION

The goal of *patch extraction* is to first find which parts of each input image are relevant for $f$.

**Patching** Since we are working with images throughout this work, we separate each of them into a grid of $n_p \times n_p$ patches. Patches have been a common way in the literature to tokenize images (Dosovitskiy et al., 2021; Tolstikhin et al., 2021; Trockman & Kolter, 2023), as they allow to preserve the locality between pixels, a key property in the vision domain. Thus, from each input $\mathbf{x}_i \in \mathbb{X}_{\mathrm{disc}}$, we obtain a set $\mathbb{P}_i := \{\mathbf{p}_{i,j} \in \mathbb{R}^{\frac{d_1}{n_p^2}}\}_{j=1}^{n_p^2}$ of $n_p^2$ patches.

**Local importance score from any feature attribution function** Then, we want to identify and select which patches of $\mathbf{x}_i$ are relevant for $f$. To do so, we rely on a *feature attribution method* $\sigma_f : \mathbb{R}^{d_1} \to [0,1]^{d_1}$, that will attribute a score to each pixel of the image in the form of a *feature attribution* map $\mathbf{s}_i = \sigma_f(\mathbf{x}_i)$. This feature attribution method can be chosen among the abundant literature on attribution-based methods. For instance, we used GradCAM (Selvaraju et al., 2017) and B-cos (Böhle et al., 2024) versions of the models in the experiments to extract feature attribution maps. More details on Bcos implementation can be found in Appendix D. By replicating the separation into patch for $\mathbf{s}_i$, we also obtain the set of saliency maps of each patch $\mathbb{S}_i = \{\mathbf{s}_{i,j} \in [0,1]^{\frac{d_1}{n_p^2}}\}_{j=1}^{n_p^2}$. From the patch-level saliency maps $\mathbb{S}_i$, we then compute a *local importance score* $v_{i,j} \in [0,1]$ for patch $j$ of image $i$ defined as

$$v_{i,j}(j, \mathbb{S}_i) = \frac{\sum_{n=1}^{\frac{d_1}{n_p^2}} s_{i,j,n}}{\sum_{m=1}^{n_p^2} \sum_{n=1}^{\frac{d_1}{n_p^2}} s_{i,m,n}}, \quad \text{such that} \quad \sum_{j=1}^{n_p^2} v_{i,j}(j, \mathbb{S}_i) = 1. \tag{1}$$

For each patch $j$ of image $i$, its local importance score $v_{i,j}$ represents the *contribution* of the patch to the model's decision, according to the feature attribution method.

**Important patches** Using $v_{i,j}$, we extract the most important patches from $\mathbb{P}_i$, by selecting those whose accumulated local importance scores, *in a decreasing order*, reaches a given *local importance threshold* $\eta_{\mathrm{local}} \in [0,1]$, to obtain $\mathbb{P}_i^* := \mathcal{E}_1(f, \mathbf{x}_i; n_p, \mathbf{s}_i, \eta_{\mathrm{local}})$.

This patch extract process $\mathcal{E}_1$ is then repeated on all images, and we extend the notation to the whole dataset $\mathbb{X}_{\mathrm{disc}}$ such that:

$$\mathbb{P}_{\mathrm{disc}}^* := \mathcal{E}_1(f, \mathbb{X}_{\mathrm{disc}}; n_p, \sigma_f, \eta_{\mathrm{local}}) = \{\mathbb{P}_i^*\}_{i=1}^{N_{\mathrm{disc}}}, \tag{2}$$

with the number of patches within each image $n_p$, the feature attribution method $\sigma_f$ and the threshold on local importance score $\eta_{\mathrm{local}}$, being hyperparameters.

### 3.1.2 PATCH EMBEDDING

From the global set of important patches $\mathbb{P}_{\mathrm{disc}}^*$, we want to summarize them into a smaller number of concepts. Since the patches still reside in a high-dimensional space $d_1$, we rely on an encoder model $g : \mathbb{R}^{d_1} \to \mathbb{R}^{d_2}$ to reduce their dimensionality in order to cluster them in a meaningful way. The choice of the encoder $g$ is vital as it should put similar patches closer together in the latent space and this closeness should be acceptable for humans.

Methods like ACE (Ghorbani et al., 2019) and CRAFT (Fel et al., 2024) utilize the network that they explain to extract a lower dimension representation for concepts. It is based on prior works (Zhang et al., 2018) showing that the Euclidean distance of the representations in the final layers of a deep Convolutional Neural Network is a good perceptual similarity metric. Following the advent of Vision Transformers and foundation models, and their performance that surpasses CNNs, it has been shown that they can be a good choice to measure perceptual similarity as well (Chan et al., 2022).

In this work, we use Dreamsim (Fu et al., 2023) as our encoder $g$, as it has shown superior performance in measuring perceptual similarities that aligns with human perception. There are different flavors of Dreamsim. The most performant combines an ensemble of DINO (Caron et al., 2021), CLIP (Radford et al., 2021), and OpenCLIP (Cherti et al., 2023) as the backbone, which makes it computationally expensive. For this work, we chose the flavor that uses OpenCLIP as backbone, since it is a good compromise between computation complexity and perceptual similarity. Using

a similar encoder for different architectures, instead of using the network we are explaining as an encoder, unifies the way we measure perceptual similarity among different architectures and allows our framework to have a consistent performance among different architectures irrespective of the performance of the model to measure the perceptual similarity of the patches.

Thus, we embed each important patch $\mathbf{p}_{i,j}^* \in \mathbb{P}_{\text{disc}}^*$ into $\mathbf{e}_{i,j}^* = g(\mathbf{p}_{i,j}^*) \in \mathbb{R}^{d_2}$ by passing them through $g$ after being reshaped to the appropriate size. This gives us our set of embedded patches $\mathbb{E}_{\text{disc}}^*$, that will be used for clustering and extracting concepts in the next stage. The patch embedding subpart can then be written as

$$\mathbb{E}_{\text{disc}}^* := \mathcal{E}_2(\mathbb{P}_{\text{disc}}^*; g) = \{\mathbf{e}_{i,j}^* = g(\mathbf{p}_{i,j}^*), \forall \mathbf{p}_{i,j}^* \in \mathbb{P}_{\text{disc}}^*\}, \tag{3}$$

where the encoder model $g$ is taken as parameter.

### 3.1.3 CONCEPTS CLUSTERING AND IMPORTANCE

From the set of important embedded patches, the last remaining subtask of concept discovery consists in summarizing them into concepts to create our concept dictionary, and linking them to actual classes learned by $f$.

**Clustering** The first step of this subtask is to cluster the embedded patches into a smaller set of concept vectors. For this, we rely on the $k$-means algorithm $\mathcal{C}$ for clustering, for its simplicity and efficiency, but any kind of clustering algorithm could be considered. We cluster the embedded patches $\mathbb{E}_{\text{disc}}^*$ found in the previous step, such that:

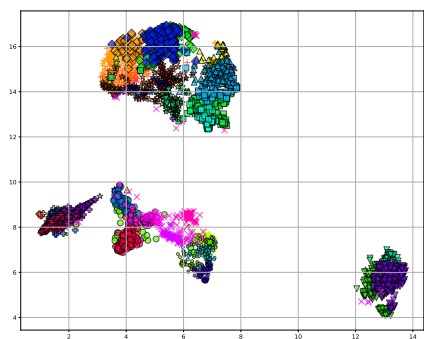

$$\mathbb{C}_{\text{disc}} := \mathcal{C}(\mathbb{E}_{\text{disc}}^*, k) = \{\mathbb{C}_n\}_{n=1}^k, \tag{4}$$

to obtain the set of $k$ clusters $\mathbb{C}_{\text{disc}}$. Figure 2 shows a UMAP (McInnes et al., 2018) plot of the embeddings of patches extracted in Dreamsim latent spaces, along with the clusters obtained and the orig-

Figure 2: UMAP of the patches embedded in the Dreamsim latent space. Each shape represents a class, and each color is a cluster of concept extracted after clustering.

inal class they belong to. We used a subset of ImageNet, described in Section 4, to extract the patches and concepts. We can see that clusters of concept can span multiple classes, which is consistent with our intuition of having concepts shared between classes. Then, we define our concepts $\mathbf{c}_n$ as the prototypes of each cluster $\mathbb{C}_n$, i.e., the average of all embedded patches $\mathbf{e}_{i,j}^* \in \mathbb{R}^{d_2}$ within each cluster, that we group into a global dictionary of concept $\mathbb{D}_{\text{disc}}$ as follows

$$\mathbb{D}_{\text{disc}} := \left\{ \mathbf{c}_n = \frac{1}{|\mathbb{C}_n|} \sum_{\mathbf{e}_{i,j}^* \in \mathbb{C}_n} \mathbf{e}_{i,j}^* \right\}_{n=1}^k . \tag{5}$$

**Importance matrix** In order to link the $k$ concepts extracted to the set of $l$ classes, we additionally compute an *importance matrix* $\mathbf{W}^{\text{disc}} \in [0, 1]^{l \times k}$. Each entry $(m, n) \in \{1, \ldots, l\} \times \{1, \ldots, k\}$ is obtained by counting the number of embedded patches in cluster $\mathbb{C}_n$ belonging to class $m$ normalized by the number of patches from images of class $m$, as follows

$$\mathbf{W}_{m,n}^{\text{disc}} := \frac{|\{\mathbf{e}_{i,j}^* \in \mathbb{C}_n | y_i = m\}|}{|\{\mathbf{e}_{i,j}^* \in \mathbb{E}_{\text{disc}}^* | y_i = m\}|} \in [0, 1]. \tag{6}$$

In other words, $\mathbf{W}_{m,n}^{\text{disc}}$ gives us the *proportion* of important patches from class $m$ that has been associated to concept $n$, such that $\forall m \in \{1, \ldots, l\}, \sum_{n=1}^k \mathbf{W}_{m,n}^{\text{disc}} = 1$. The concept clustering part $\mathcal{D}_3$ can then be summarized as

$$\mathbb{D}_{\text{disc}}, \mathbf{W}^{\text{disc}} := \mathcal{D}_3(\mathbb{E}_{\text{disc}}^*; k) \tag{7}$$

$$:= (\mathcal{D}_3 \circ \mathcal{E}_2 \circ \mathcal{E}_1)(f, \mathbb{X}_{\text{disc}}; n_p, \sigma_f, \eta, g, k), \tag{8}$$

with $k$ the number of concepts considered in the clustering. The extracted concept dictionary $\mathbb{D}_{\text{disc}}$ and the importance matrix $\mathbf{W}^{\text{disc}}$ are the outputs that will be used during *concept assignment* $\mathcal{A}$.

## 3.2 CONCEPT ASSIGNMENT

At testing time, the *concept assignment* algorithm $\mathcal{A}$, explains the decision of the model $f$ under investigation from the concept dictionary $\mathbb{D}_{\text{disc}}$ and the class-concept importance matrix $\mathbf{W}^{\text{disc}}$ extracted in the first phase. The goal of concept assignment is to explain the prediction of $f$ on new unseen data. We first describe the overall algorithm for a single test sample $\mathbf{x}_{\text{test}} \in \mathbb{R}^{d_1}$, i.e. for *local interpretation*, and then its extension to a whole dataset $\mathbb{X}_{\text{test}} = \{\mathbf{x}_i^{\text{test}} \in \mathbb{R}^{d_1}\}_{i=1}^{N_{\text{test}}}$, i.e. for *global interpretation*. Here again, we decompose the assignment task into three subtasks (functions), with the first two ones being shared with the concept discovery, namely *patch extraction* $\mathcal{E}_1$, *patch embedding* $\mathcal{E}_2$, and then *concept assignment* $\mathcal{A}_3$. We detail these functions below.

### 3.2.1 PATCH EXTRACTION AND EMBEDDING

To be able to find and assign concepts in $\mathbf{x}_{\text{test}}$ to the ones discovered in our concept dictionary $\mathbb{D}_{\text{disc}}$, we need first to extract and embed patches, and thus follow the same process of decomposing $\mathbf{x}_{\text{test}}$ into $n_p \times n_p$ patches. However, to alleviate the requirement of feature attribution map at test time and to avoid discarding useful information, we extract all patches and postpone the selection of important ones in the next step. One should note that this is equivalent to considering a feature attribution function $\mathbb{1} : \mathbb{R}^{d_1} \to \{1\}^{d_1}$ that assigns the score of 1 to every pixel, along with a local importance threshold $\eta_{\text{local}} = 1$. The patch extraction during concept assignment can then be written as

$$\mathbb{P}_{\text{test}}^* = \mathbb{P}_{\text{test}} := \mathcal{E}_1(f, \mathbf{x}_{\text{test}}; n_p, \mathbb{1}, 1) = \{\mathbf{p}_{\text{test},j} \in \mathbb{R}^{\frac{d_1}{n_p^2}}\}_{j=1}^{n_p^2}. \tag{9}$$

Then, we apply the same patch embedding process, using the same encoder $g$ as in the concept discovery phase, as follows

$$\mathbb{E}_{\text{test}} := \mathcal{E}_2(\mathbb{P}_{\text{test}}; g) = \{\mathbf{e}_{\text{test},j} = g(\mathbf{p}_{\text{test},j}) \in \mathbb{R}^{d_2}, \forall \mathbf{p}_{\text{test},j} \in \mathbb{P}_{\text{test}}\}, \tag{10}$$

to obtain our set of embedded patches $\mathbb{E}_{\text{test}}$, lying in the same space $\mathbb{R}^{d_2}$ as our concepts.

### 3.2.2 CONCEPT ASSIGNMENT

Now, the goal of concept assignment is to find *important concepts* within $\mathbf{x}_{\text{test}}$, i.e. concepts useful for prediction. Given an embedded patch $\mathbf{e}_{\text{test},j} \in \mathbb{E}_{\text{test}}$, we compute the Euclidean distances to all concepts in our dictionary $\mathbf{c}_n \in \mathbb{D}_{\text{disc}}$, to find the closest one $\hat{n}$ such that

$$\hat{n} := \arg \min_{\substack{n \in \{1,\ldots,k\}, \\ \mathbf{c}_n \in \mathbb{D}_{\text{disc}}}} \|\mathbf{e}_{\text{test},j} - \mathbf{c}_n\|_2. \tag{11}$$

Then, we consider both the embedded patch $\mathbf{e}_{\text{test},j}$ and its assigned concept $\mathbf{c}_{\hat{n}}$ as important for prediction, if they fulfill both of the following conditions:

- $\mathbf{e}_{\text{test},j}$ resides within the *hypersphere* of cluster $\mathbb{C}_{\hat{n}}$, whose radius $R_{\hat{n}} > 0$ is defined by the distance to its furthest embedded patch:

$$\|\mathbf{e}_{\text{test},j} - \mathbf{c}_{\hat{n}}\|_2 \leq \max_{\mathbf{e}_{i,j}^* \in \mathbb{C}_{\hat{n}}} \|\mathbf{e}_{i,j}^* - \mathbf{c}_{\hat{n}}\|_2 = R_{\hat{n}}, \tag{12}$$

- the *number of important patches* within cluster $\mathbb{C}_{\hat{n}}$ and associated to the predicted class $\hat{y} = f(\mathbf{x}_{\text{test}})$ is higher than a given *global importance threshold* $\eta_{\text{global}} > 0$:

$$\mathbf{W}_{\hat{y},\hat{n}}^{\text{disc}} \geq \frac{\eta_{\text{global}}}{|\{\mathbf{e}_{i,j}^* \in \mathbb{E}_{\text{disc}}^* | y_i = \hat{y}\}|}. \tag{13}$$

We reproduce this process for each embedded patch $\mathbf{e}_{\text{test},j} \in \mathbb{E}_{\text{test}}$, and group important patches and which concepts they are assigned to, respectively into $\mathbb{E}_{\text{test}}^*$ and $\mathbb{C}_{\text{test}}^*$, such that

$$\mathbb{E}_{\text{test}}^* := \left\{ \mathbf{e}_{\text{test},j} \in \mathbb{E}_{\text{test}} \mid (\|\mathbf{e}_{\text{test},j} - \mathbf{c}_{\hat{n}}\|_2 \leq R_{\hat{n}}) \wedge \left( \mathbf{W}_{\hat{y},\hat{n}}^{\text{disc}} \geq \frac{\eta_{\text{global}}}{|\{\mathbf{e}_{i,j}^* \in \mathbb{E}_{\text{disc}}^* | y_i = \hat{y}\}|} \right) \right\} \tag{14}$$

$$\mathbb{C}_{\text{test}}^* := \left\{ \mathbb{C}_{\hat{n}}^{\text{test}} = \left\{ \mathbf{e}_{\text{test},j} \in \mathbb{E}_{\text{test}}^* \mid \hat{n} = \arg \min_{\substack{n \in \{1,\ldots,k\}, \\ \mathbf{c}_n \in \mathbb{D}_{\text{disc}}}} \|\mathbf{e}_{\text{test},j} - \mathbf{c}_n\|_2 \right\} \right\}_{\hat{n}=1}^k \tag{15}$$

We additionally extract a relevance score $\mathbf{w}^{\text{test}} \in [0,1]^k$ for each concept, defined as the proportion of important patches assigned to each concept, as follows

$$\mathbf{w}_{\hat{n}}^{\text{test}} := \frac{|\mathbb{C}_{\hat{n}}^{\text{test}}|}{|\mathbb{E}_{\text{test}}^*|}. \tag{16}$$

Finally, the concept assignment for a single test sample can be summarized as

$$\mathbb{E}_{\text{test}}^*, \mathbb{C}_{\text{test}}^*, \mathbf{w}^{\text{test}} := \mathcal{A}_3(\mathbb{E}_{\text{test}}; \mathbb{D}_{\text{disc}}, \mathbb{C}_{\text{disc}}, \mathbb{E}_{\text{disc}}^*, \eta_{\text{global}}) \tag{17}$$

$$:= (\mathcal{A}_3 \circ \mathcal{E}_2 \circ \mathcal{E}_1)(f, \mathbf{x}_{\text{test}}; n_p, \mathbb{1}, 1, g, \mathbb{D}_{\text{disc}}, \mathbb{C}_{\text{disc}}, \mathbb{E}_{\text{disc}}^*, \eta_{\text{global}}). \tag{18}$$

In the next section, we describe the extension to a *test dataset* $\mathbb{X}_{\text{test}}$ for global interpretation.

### 3.2.3 GLOBAL INTERPRETATION

Similarly to the concept discovery phase, we extend the notation of patch extract $\mathcal{E}_1$ to process the whole dataset $\mathbb{X}_{\text{test}} = \{\mathbf{x}_i^{\text{test}} \in \mathbb{R}^{d_1}\}_{i=1}^{N_{\text{test}}}$, such that

$$\mathbb{P}_{\text{test}}^* = \mathbb{P}_{\text{test}} := \mathcal{E}_1(f, \mathbb{X}_{\text{test}}; n_p, \mathbb{1}, 1) = \{\mathbb{P}_i^{\text{test}} := \mathcal{E}_1(f, \mathbf{x}_i^{\text{test}}; n_p, \mathbb{1}, 1)\}_{i=1}^{N_{\text{test}}}. \tag{19}$$

From there, as for a single test sample, we extract the set of all embedded patches $\mathbb{E}_{\text{test}} := \mathcal{E}_2(\mathbb{P}_{\text{test}}; g) = \{\mathbf{e}_{i,j} = g(\mathbf{p}_{i,j}) \in \mathbb{R}^{d_2}, \forall \mathbf{p}_{i,j} \in \mathbb{P}_{\text{test}}\}$ from all images, and find important patches $\mathbb{E}_{\text{test}}^*$ and their assigned concepts $\mathbb{C}_{\text{test}}^*$ that fulfill both conditions described in Equation (12) and Equation (13) from all the embedded patches. Instead of a single vector of relevance scores, we extract a relevance matrix $\mathbf{W}^{\text{test}} \in [0,1]^{l \times k}$, that aggregates the relevance scores of each concept for each predicted class $\hat{m} = f(\mathbf{x}_i^{\text{test}})$ of samples $\mathbf{x}_i^{\text{test}} \in \mathbb{X}_{\text{test}}$, defined as follows

$$\mathbf{W}_{\hat{m},\hat{n}}^{\text{test}} := \frac{|\{\mathbf{e}_{i,j} \in \mathbb{C}_{\hat{n}}^{\text{test}} \mid f(\mathbf{x}_i^{\text{test}}) = \hat{m}\}|}{|\{\mathbf{e}_{i,j} \in \mathbb{E}_{\text{test}}^* \mid f(\mathbf{x}_i^{\text{test}}) = \hat{m}\}|}, \tag{20}$$

where $\mathbb{C}_{\hat{n}}^{\text{test}} \in \mathbb{C}_{\text{test}}^*$ corresponds the set of embedded patches from $\mathbb{E}_{\text{test}}^*$ assigned to cluster $\mathbb{C}_{\hat{n}}$. Concept assignment for the whole dataset $\mathbb{X}_{\text{test}}$ can then be summarized as

$$\mathbb{E}_{\text{test}}^*, \mathbb{C}_{\text{test}}^*, \mathbf{W}^{\text{test}} := (\mathcal{A}_3 \circ \mathcal{E}_2 \circ \mathcal{E}_1)(f, \mathbb{X}_{\text{test}}; n_p, \mathbb{1}, 1, g, \mathbb{D}_{\text{disc}}, \mathbb{C}_{\text{disc}}, \mathbb{E}_{\text{disc}}^*, \eta_{\text{global}}). \tag{21}$$

Finally, to explain a classifier $f$, we can analyze the important patches extracted for each sample $\mathbb{P}_{\text{test}}$ if we want to understand the predictions locally, analyze the global class-concept relevance matrix $\mathbf{W}^{\text{test}}$ to understand their relationships, and also interpret the meaning of each concept in our dictionary $\mathbb{D}_{\text{disc}}$ by visualizing the patches within each cluster. In the next section, we present experimental results, both quantitative and qualitative, when extracting concepts to explain pretrained classifiers $f$ after concept assignment. We describe the experimental settings considered below.

## 4 EXPERIMENTAL RESULTS

In this section, we study CAN through various numerical experiments and compare it with two other post-hoc concept extraction methods, namely ACE (Ghorbani et al., 2019) and MCD (Vielhaben et al., 2023). We consider two datasets, ImageNet (Deng et al., 2009) and CUB (Wah et al., 2011). For the former, we extract concepts on a subset of ten classes that roughly aligned with CIFAR-10 classes (Krizhevsky et al., 2009) as introduced in (Vielhaben et al., 2023), and on a subset of ten random classes for the latter. Results on CUB dataset can be found in Appendix C. We have selected Resnet (He et al., 2016), and ViT$_C$ (Xiao et al., 2021) as architectures to showcase the versatility of CAN. Indeed, our framework can be applied to either CNNs (LeCun et al., 1989) or ViTs (Dosovitskiy et al., 2021), as long as a method to extract feature attribution maps is available. Typically, we rely here on GradCAM (Selvaraju et al., 2017) and B-cos (Böhle et al., 2024) versions of the models to extract the salient regions of input images. Furthermore, we also include results using all ImageNet classes with CAN.

### 4.1 QUANTITATIVE RESULTS

**Faithfulness** Faithfulness measures the importance of concepts in two complementary ways. First how important the concepts are to the model by measuring the drop in accuracy when removing concepts, a setting called *Smallest Destroying Concept (SDC)*, and, second, how accurate the model is when given images representing only few important concepts as input, called *Smallest Sufficient Concept (SSC)* (Ghorbani et al., 2019). In SDC, we remove all the pixels belonging to a concept, from the most important to the least important one. The importance of each concept is defined differently for each method considered. ACE uses the *TCAV score* (Kim et al., 2018), MCD uses *local concept importance* (Vielhaben et al., 2023), whereas we look at entries in our global relevance matrix $\mathbf{W}^{\text{test}}$, computed during concept assignment. For SSC, we start with a black image and add concepts following their order of importance, from highest to lowest, and then measure the accuracy of the model at each step. As concepts can have different sizes depending on the methods, we plot faithfulness with respect to percentage of concepts' pixels, similarly to Vielhaben et al. (2023). Results from Figure 3 show that concepts discovered by CAN are generally more faithful to the

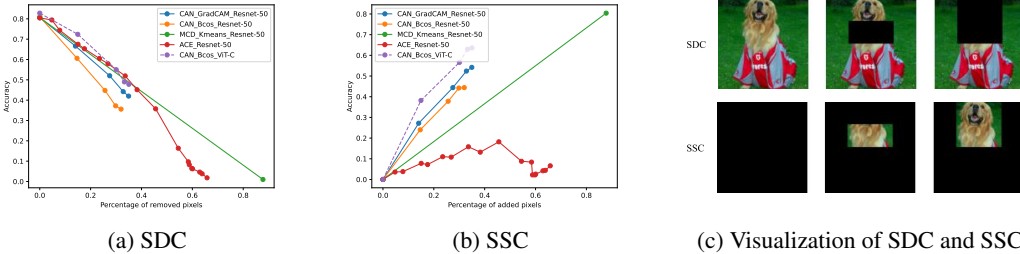

|          (a) SDC          |          (b) SSC          |    (c) Visualization of SDC and SSC    |

Figure 3: Drop in accuracy (a) and increase in accuracy (b) when adding or removing concepts one by one, depending on the accumulated percentage of pixels, for concepts extracted from CIFAR-10 classes of Imagenet. (c) Visualization of patches associated to concepts removed (for SDC) or added (SSC) using CAN on an example image.

model. On SDC (and resp. SSC) experiments, we can see that removing (resp. adding) concepts with lower size leads to a higher decrease (resp. increase) of accuracy. Notably, MCD assigns on average more than 87% of pixels of each image to only a single concept. This behavior prevents from dividing input images into multiple concepts. Furthermore, ACE, by-design, finds concepts per class. We also measure the faithfulness of CAN on a Resnet-50 and ViT$_C$ for all classes in ImageNet, with results shown in Figure 4. As in the previous case, using Bcos as a feature attribution method allows for finding more faithful concepts to remove, whereas in SSC, on the other hand, GradCAM finds more important concepts to add first.

**Conciseness** Conciseness can be defined as the number of concepts required to explain a class (Vielhaben et al., 2023; Parekh et al., 2021). In practice, we are interested in concise explanations, i.e., using fewer concepts to explain the model. However, very low values of conciseness are not desirable, as we still want a detailed explanation including multiple concepts (Vielhaben et al., 2023).

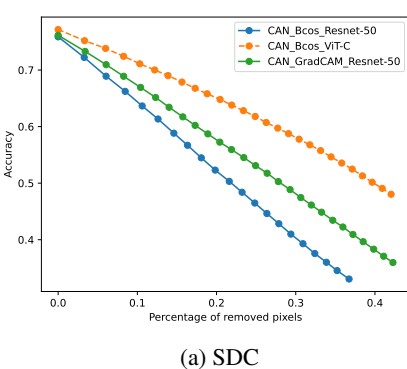
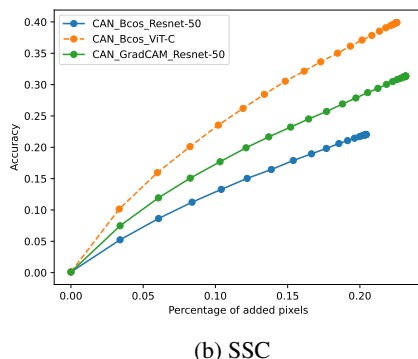

(a) SDC

(b) SSC

Figure 4: Drop in accuracy (a) and increase in accuracy (b) when adding or removing concepts one by one, depending on the accumulated percentage of pixels, for concepts extracted from all classes of Imagenet.

Table 2: **(a)** Comparison of conciseness of different post-hoc concept extraction methods, depending on the architectures. **(b)** Consistency (accuracies in %) of concept dictionary, depending on the number of clusters (i.e. number of concepts) considered in concept discovery, for Resnet-50 and $\text{ViT}_C$, on ImageNet.

(a) Conciseness of explanation

| Arch. | Method | Conciseness |
|-------|--------|-------------|
| Resnet-50 | MCD | 1 |
| | ACE | 11.6 |
| | CAN - GradCAM | 2.8 |
| | CAN - Bcos | 2.8 |
| $\text{ViT}_C$ | CAN - Bcos | 2.8 |

(b) Consistency of concept dictionary

| Nb concepts | Resnet-50 | $\text{ViT}_C$ |
|-------------|-----------|-----------------|
| 1000 | 8.20 | 7.12 |
| 2000 | 9.72 | 8.61 |
| 3000 | 10.22 | 9.64 |
| 4000 | 10.49 | 10.04 |
| 5000 | 10.63 | 10.92 |

From Table 2a, we can see that CAN uses on average 2.8 concepts for each class, which lies between values of MCD and ACE, 1 and 11.6 concepts per class respectively.

## 4.2 QUALITATIVE RESULTS

The Concept Assignment algorithm can be used to extract *where* a given concept is located and *what* is the meaning of that concept, for a given image. The meaning of the concepts can be inferred from the closest patches to the center of the concept's cluster. Here we have shown the 5 closest patches to each concept for this purpose. Figure 5 shows the visualization of the concept assignment for an image from the class *Airliner* for different architectures and feature attribution methods. It can be seen that the concepts extracted by CAN are semantically similar for different architectures and feature attribution methods.

## 4.3 CONCEPT DICTIONARY CONSISTENCY

Since our method allows for extracting concepts among a dictionary shared for all classes, we propose a novel experimental protocol to evaluate the *consistency* of the concept dictionary, over a test dataset $\mathbb{X}_{\text{test}}$. Given a number $k$ of clusters, we obtain our corresponding dictionary of concepts $\mathbb{D}_{\text{disc}}$ from *concept discovery*, and find the important patches $\mathbb{E}_{\text{test}}^*$, their assigned concepts $\mathbb{C}_{\text{test}}^*$ and the class-concept relevance matrix $\mathbf{W}^{\text{test}}$ from *concept assignment* on $\mathbb{X}_{\text{test}}$. Then, for each image $\mathbf{x}_i^{\text{test}} \in \mathbb{X}_{\text{test}}$, we compute a *concept distance feature vector* $\phi_i$ from the sum of Euclidean distance of all the important patches $\{\mathbf{e}_{i,j} \in \mathbb{E}_{\text{test}}^*\}_{j=1}^{n_p^2}$ of this image to all concepts $\mathbf{c}_n \in \mathbb{D}_{\text{disc}}$, weighted by their class-concept relevance in $\mathbf{W}^{\text{test}}$, as follows:

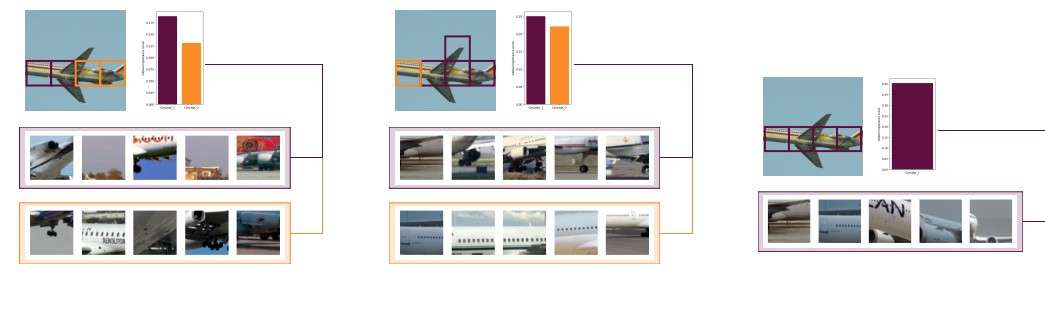

(a) CAN - GradCAM - Resnet-50      (b) CAN - Bcos - Resnet-50      (c) CAN - Bcos - ViT$_C$

Figure 5: Visualization of the extracted patches after Local Concept Assignment for a sample image from class *Airliner* for two architectures, Resnet-50 and ViT$_C$, and GradCAM and Bcos as feature attribution methods.

$$\phi_i := \sum_{j=1}^{n_p^2} \sum_{n=1}^{k} \mathbf{W}_{\hat{m},k}^{\text{test}} ||\mathbf{e}_{i,j} - \mathbf{c}_k||_2, \quad \text{where} \quad \hat{m} = f(\mathbf{x}_i^{\text{test}}). \tag{22}$$

Finally, we train a simple decision tree classifier on a random training subset of $\{\phi_i\}_{i=1}^{N_{\text{test}}}$, and evaluate its accuracy on the remaining test subset. The *consistency* of the concept dictionary is then defined as the accuracy of the classifier on the test subset. We present results in Table 2b of consistency for different number $k$ of concepts in our dictionary, on ImageNet. We can see that increasing the value of $k$ improves the consistency, as we are introducing more information in our feature vector. However, since we are also interested in having a *concise* dictionary, we recommend selecting $k$ where the consistency starts to plateau.

## 5 CONCLUSION

To summarize, we present a novel post-hoc concept-based interpretability method, CAN, that can be applied to arbitrary visual classifiers. CAN uses attribution map information as an intermediate signal. Through patching and clustering in the embedding space of a fixed encoder it extracts a concept dictionary using this intermediate signal. Through extensive experiments spanning multiple datasets, architectures and attribution algorithms, we demonstrated the versatility of our method and showed its ability to generate highly faithful and concise interpretations. Moreover, CAN is also capable to provide a holistic understanding with a shared concept dictionary for all classes that can easily scale even to the whole ImageNet dataset. Future works concern the association of patches-based concepts with textual concept descriptions, and the extension of this framework to non-visual modalities.

### REPRODUCIBILITY STATEMENT

Throughout the paper, we made sure that all our experiments were fully reproducible, describing in details all datasets, classes and architectures considered in Section 4, and checkpoints and hyperparameters in Appendix E.

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

## A  ALGORITHMS

The pseudo-code of different algorithms developed for CAN can be found here. Algorithm 1 is used to extract concepts as described in Section 3.1. Algorithm 2 is used to assign the extracted concepts locally to a single image. Finally, Algorithm 3 is used to explain the model globally.

---

**Algorithm 1** Concept Discovery $\mathcal{D}$

---

**Inputs:** model to explain $f$, concept discovery set $\mathbb{X}_{\text{disc}}$, number of patches $n_p$, feature attribution function $\sigma_f$, local importance threshold $\eta_{\text{local}}$, perceptual similarity encoder $g$, number of clusters $k$

$\mathbb{P}^*_{\text{disc}} \leftarrow \varnothing$

**for** $\forall \mathbf{x}_i \in \mathbb{X}_{\text{disc}}$ **do**

$\quad \mathbf{s}_i := \sigma_f(\mathbf{x}_i)$

$\quad \mathbb{P}^*_i := \mathcal{E}_1(f, \mathbf{x}_i; n_p, \mathbf{s}_i, \eta_{\text{local}})$                     {Extract important patches}

$\quad \mathbb{P}^*_{\text{disc}} \leftarrow \mathbb{P}^*_{\text{disc}} \cup \mathbb{P}^*_i$

**end for**

$\mathbb{E}^*_{\text{disc}} := \mathcal{E}_2(\mathbb{P}^*_{\text{disc}}; g)$                        {Embed patches into lower dimensional space}

$\mathbb{D}_{\text{disc}}, \mathbf{W}^{\text{disc}} := \mathcal{D}_3(\mathbb{E}^*_{\text{disc}}, k)$                      {Cluster embedded patches}

**Outputs:** concept dictionary $\mathbb{D}_{\text{disc}}$, importance matrix $\mathbf{W}^{\text{disc}}$.

---

**Algorithm 2** Concept Assignment $\mathcal{A}$ - local interpretation

---

**Inputs:** model to explain $f$, test sample $\mathbf{x}_{\text{test}}$, number of patches $n_p$, global importance threshold $\eta_{\text{global}}$, perceptual similarity encoder $g$, concept dictionary $\mathbb{D}_{\text{disc}}$, concept clusters $\mathbb{C}_{\text{disc}}$, important embedded patch $\mathbb{E}^*_{\text{disc}}$

$\mathbb{P}_{\text{test}} := \mathcal{E}_1(f, \mathbf{x}_{\text{test}}; n_p, \mathbb{1}, 1)$                      {Extract all patches}

$\mathbb{E}_{\text{test}} := \mathcal{E}_2(\mathbb{P}_{\text{test}}; g)$                    {Embed patches into lower dimensional space}

$\mathbb{E}^*_{\text{test}}, \mathbb{C}^*_{\text{test}}, \mathbf{w}^{\text{test}} = \mathcal{A}_3(\mathbb{E}_{\text{test}}; \mathbb{C}_{\text{disc}}; \mathbb{E}^*_{\text{disc}}, \eta_{\text{global}})$     {Find closest clusters to embedded patches}

**Outputs:** Important patches $\mathbb{E}^*_{\text{test}}$, assigned concepts $\mathbb{C}^*_{\text{test}}$, relevance score $\mathbf{w}^{\text{test}}$

---

**Algorithm 3** Concept Assignment $\mathcal{A}$ - global interpretation

---

**Inputs:** model to explain $f$, test dataset $\mathbb{X}_{\text{test}}$, number of patches $n_p$, global importance threshold $\eta_{\text{global}}$, perceptual similarity encoder $g$, concept dictionary $\mathbb{D}_{\text{disc}}$, concept clusters $\mathbb{C}_{\text{disc}}$, important embedded patch $\mathbb{E}^*_{\text{disc}}$

$\mathbb{P}_{\text{test}} \leftarrow \varnothing$

**for** $\forall \mathbf{x}_i^{\text{test}} \in \mathbb{X}_{\text{test}}$ **do**

$\quad \mathbb{P}_i^{\text{test}} := \mathcal{E}_1(f, \mathbf{x}_i^{\text{test}}; n_p, \mathbb{1}, 1)$                      {Extract all patches}

$\quad \mathbb{P}_{\text{test}} \leftarrow \mathbb{P}_{\text{test}} \cup \mathbb{P}_i^{\text{test}}$

**end for**

$\mathbb{E}_{\text{test}} = \mathcal{E}_2(\mathbb{P}_{\text{test}}; g)$                     {Embed patches into lower dimensional space}

$\mathbb{E}^*_{\text{test}}, \mathbb{C}^*_{\text{test}}, \mathbf{W}^{\text{test}} = \mathcal{A}_3(\mathbb{E}_{\text{test}}; \mathbb{C}_{\text{disc}}, \mathbb{E}^*_{\text{disc}}, \eta_{\text{global}})$   {Find closest clusters to embedded patches}

**Outputs:** Important patches $\mathbb{E}^*_{\text{test}}$, assigned concepts $\mathbb{C}^*_{\text{test}}$, relevance matrix $\mathbf{W}^{\text{test}}$

---

## B  CONCEPT ASSIGNMENT

A high-level overview of concept-assignment algorithm introduced in Section 3.2 is shown in Figure 6.

## C  FAITHFULNESS AND CONCISENESS OF A MODEL TRAINED ON CUB DATASET

Figure 7 shows the faithfulness of CAN to explain the Resnet-50 trained on CUB dataset, Bcos. Like the case with Imagenet, MCD assigns a large portion of the input image to only one concept

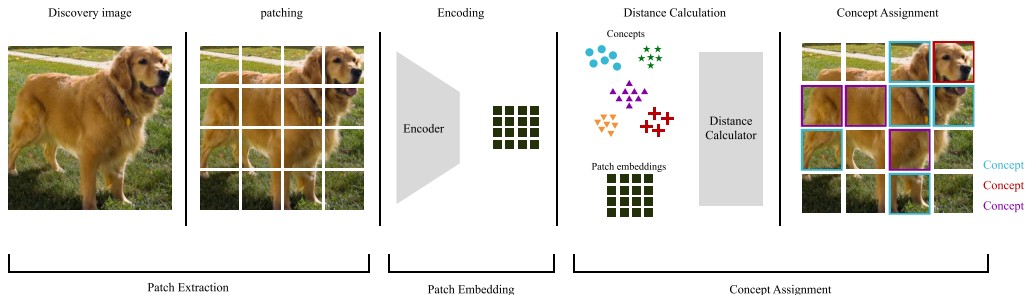

Figure 6: A high-level overview of concept Assignment in CAN framework. A single image is divided into $n_p \times n_p$ patches. These patches are passed through an encoder and the distance between each patch and each concept is calculated. Finally, the patches are assigned to the closest concept that has been found in Concept Discovery.

| Arch. | Method | Conciseness |
|---|---|---|
| Resnet-50 | MCD | 1 |
| | CAN - Bcos | 4 |

Table 3: Conciseness of explanation of MCD and CAN to explain Resnet-50.

that is not favorable, whereas CAN assigns multiple concepts to each class that can lead to more fine-grained concepts. The conciseness of CAN and MCD is also reported in Table 3.

## D  ADOPTING BCOS TO CAN

Bcos networks (Böhle et al., 2024) are inherently explainable by design. However, they possess a unique property, making them a perfect alternative to replace the feature attribution method in CAN. In a Bcos network, the model's operations can be replaced by a linear transform $\mathbf{W}_{1 \to L} \in \mathbb{R}^{l \times C \times H \times W}$ that summarizes the operations from the first layer to the last layer, with $l$ the number of classes. To adopt Bcos to CAN we can safely use the spatial contribution map corresponding to prediction $\hat{y} := f(\mathbf{x}_i)$, computed from $\mathbf{W}_{1 \to L}$ as our feature attribution map $\mathbf{s}_i$ of input $\mathbf{x}_i$:

$$\mathbf{s}_i := \sum_{c=1}^{C} \left( [\mathbf{W}_{1 \to L}(\mathbf{x}_i)]_{\hat{y}}^{\top} \odot \mathbf{x}_i \right)_c. \tag{23}$$

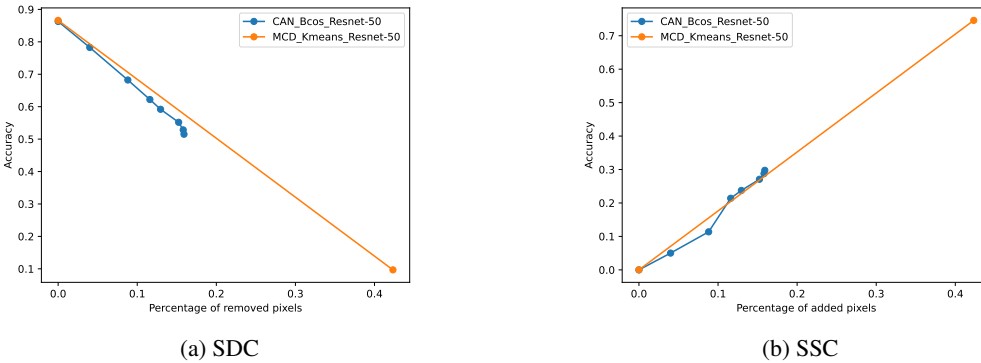

(a) SDC                    (b) SSC

Figure 7: Drop in accuracy (a) and increase in accuracy (b) when adding or removing concepts one by one, depending on the accumulated percentage of pixels, for concepts extracted from 10 random classes of CUB.

# E   IMPLEMENTATION DETAILS

For this work, we used the pretrained Resnet-50 from the Torchvision library, and the pre-trained Bcos versions of Resnet-50 and ViT$_C$ provided by the authors in their official github repository (`https://github.com/B-cos/B-cos-v2?tab=readme-ov-file`). For experiments over CUB dataset, we changed the classifier head of the model to the appropriate dataset size, 200, and retrained the model following the procedure defined to train Bcos networks. We chose the number of patches, $n_p$, equal to 4, hence dividing each image into 16 patches. $\eta_{local}$ is set to 0.5 so that $\sum_n v_{i,j}(j, \mathbb{S}_i) \geq 0.5$, and $\eta_{global}$ is set to 2.

