# OpenReview forum: "Conceptualize Any Network: A Concept Extraction Framework for Holistic Interpretability of Image Classifiers"
_ICLR.cc/2025/Conference — ICLR 2025 Conference Withdrawn Submission_

### Official Review · Reviewer_54po · 2024-10-25

**Soundness:** 3
**Presentation:** 3
**Contribution:** 2
**Rating:** 3
**Confidence:** 4

**Summary:**

The authors propose a new model agnostic concept extraction framework which can provide both local and global explanations of concepts learned by image classifiers. Their method is built upon the concept of embedding important image features into a common concept space for all classes. They first break an image into patches and use an attribution method to find an importance of each patch. Then, the highest importance patches are embedded. This is done for a selection of N image samples from all classes to form a concept embedding space. Then, local and global concept explanations are made by comparing all the patches in the subject images to those important patch concepts in the embedding space.

**Strengths:**

The approach is interesting and promising. There is much value in a concept extraction framework which can conceptualize knowledge on a local and global scale, especially if it does so agnostically.

The approach is logical, simple, and easy to understand. I believe this to be a smart way to solve the proposed problem.

I also believe the contribution, if it yielded better results would be valuable to the literature.

**Weaknesses:**

Multiple hyperparameters are introduced but they are never given explicit values, and no ablation studies were performed to indicate their selection process.

One qualitative figure is not enough to be convincing that this is method is effective. The example in Figure 5 is also not entirely convincing. I would need to see a large selection of examples to make a proper evaluation of this method’s performance.

The results in Table 2a and 2b are not convincing. Table 2a seems arbitrary in terms of what level of conciseness is desirable. The low accuracies in Table 2b cast doubt on the effectiveness of this method at extracting concepts.

**Questions:**

How is the patch size chosen? Different ViT models would likely benefit from a patch size that respects their internal parameters. What determines if a patch size is small enough to capture a concept such as the wing of a bird when the entire bird may be covered by one patch if the patch size is too large.

This question extends to many hyperparameters introduced in the paper such as: local importance threshold and global importance threshold. How were the correct values for these chosen and what are they?

Consistency scores in table 2a seems low. Why are higher consistencies not achieved?

Are better qualitative results than those in Figure 5 achievable? Why is there only one qualitative result presented?

---

### Official Review · Reviewer_KQve · 2024-10-30

**Soundness:** 1
**Presentation:** 1
**Contribution:** 1
**Rating:** 1
**Confidence:** 5

**Summary:**

This paper presents a method to extract visual patches, called concepts, important for the prediction made by a given model. The extracted visual patches forms a dictionary whose visual patches are shared among all classes. To do so, the proposed method has two stages, concept discovery and concept assignment. In the concept discovery stage, considering a predefined patch size, the work firstly collects important patches from all images using an attribution method.  Secondly, to create the dictionary of concepts, it applies k-means clustering on embedding of patches produced by an external model. In the concept assignment stage, given a test image and its prediction, the method obtains important patches that can explain the made prediction. To do so, the concepts which are closest to the patch of images as well as have higher relation score with the predicted class are extracted from the concept dictionary. The described procedure is used for local interpretability task. Additionally, this procedure per image can be repeated over all test images and obtain important patches from the test set to conduct a global interpretability task. Since, the concept discovery step is independent of the given base model, the work states that the proposed framework can be applied on any network.

This paper should be rejected because (1) the points stated as contributions are either wrong or have been already introduced in the literature (2) lack of proper discussion in the related work section w.r.t the similar works (3) there are mistakes in introducing math notations (4) The reproducibility of the work is problematic since the evaluation protocol have not been explained properly (5) There is no analysis to support the claims mentioned in the work such as scalability of the works to large datasets and visualization analysis from CNNs and Vision Transformers.

**Strengths:**

One of the strength points of the work might be providing local and global explainability using concept-based approach instead of only explaining the prediction using attribution method that highlights input features. Another point is considering a vision transformer in its evaluation phase.

**Weaknesses:**

- The work presents a list of three contributions. However, I would argue these.
Firstly,  an extensive validation can not be listed as a contribution since each method must be evaluated in a proper way to show the effectiveness of its approach.
Secondly, the idea of a dictionary of visual patches shared among classes have already been introduced in the literature. I would like to suggest the work [1] which provides a quite similar approach to this paper as [1] provides a dictionary of visual patches that are class-specific or class-shared.
Thirdly, while the work stresses the versatility of its approach to any architecture, it does not discuss this characteristics w.r.t. to the similar works in the literature, properly. I would agree that ACE and SHAP have been evaluated only on CNNs. I think it should be taken into account that by the time of their publications, the literature were focusing on CNNs. The methodology of these methods consider internal representations as input. Therefor, from theoretical point of view they can be applied on Vision Transformers with proper modification. Having said that, to support the claim made by the work, it needs to discuss whether these methods have specific characteristics that tailor their methodology to the architecture of CNNs.

- The work lacks on covering recent related works such as [1] and [2], or even similar old works such as [3].

- Given the statement of the work in line 027 regarding scalability of the proposed frameworks which has not been achieved before, I would like suggesting again the work [1] and [4]. I agree with the valid observation made by this work that says ACE and ShAP can not be scaled to  large datasets due to their expensive computational cost. Having said that, the proposed work and ACE both rely on the K-means clustering approach. Therefore, there is a lack of analysis regarding the execution time or computational complexity of the work when it is scaled to the large dataset.

- It is not clear very well what is considered as concept in the paper. From lines 257 and 265 it can be understood that each cluster of visual patches is considered as a concept. However, the lines 305, 398, and 399, “important concepts within x_test”, consider visual patches as concepts.  Moreover, there is no consistency among utilized terminologies in the paper which makes understanding the text problematic. For example, post-hoc explainability, post-hoc interpretation, concept based interpretability, post-hoc concept extraction, post-hoc concept-based interpretability.

- The mathematics notations have not been introduced in a proper location. For example, $l$ is used in the beginning of section 3.1, while it is introduced later in section 3.1.3. The goal of providing equations is explaining the difficult concepts. However, equation 1 is so complex to understand. By taking a close look, it seems it simply computes the normalized summation of elements in each patch. Moreover, while it introduces $s_{i,j}$ as the patch $j$ from the image $i$, the notation $s_{i,j,n}$ is not explained.   Moreover, $n^2_p$ is not introduced. While $n$ is used to count number of patches, it is also used to count the number of clusters. There is the same thing for notations $i$ and $j$. They utilized for multiple components (Eqs. 1 and 5). Moreover, $\hat{n}$ is used for both the predicted concept and the counter in Eq. 15.

- The proposed method contains multiple hyper-parameters that effects on creating the concept dictionary. However, excepting the hyper-paramter $k$ (the number of clusters), there is no ablation study on other parameters to investigate the sensitivity of the methods on the hyper-parameters. For example, the method considers only uncovered window for extracting the patches. However, it is highly possible that a highlighted visual part might be divided among different patches.

- The labels in Fig 5 is not visible very well.

- Since, the method emphasizes that the discovered concepts are shared among classes, a proper evaluation w.r.t. this aspect would be needed with other methods provide concepts shared among classes [1].

- The explanation of evaluation protocol is not complete which makes reproducibility of the work problematic. For example, while the proposed work has two stages, concept discovery and concept assignment, ACE has only one stage which is  concept discovery. Therefore, it is not clear how ACE has been applied on the test set, or even whether all the methods have been fed with the same set of the images in the evaluation phase or not.

[1] T. Meynen, et. al. "Interpreting Convolutional Neural Networks by Explaining Their Predictions,(ICIP 2023)

[2] Kowal et. al. "Visual Concept Connectome (VCC): Open World Concept Discovery and their Interlayer Connections in Deep Models." (CVPR 2024).

[3] Li et al. “Mining mid-level visual patterns with deep cnn activations,” International Journal of Computer Vision, vol. 121, no. 3, pp. 344–364, 2017.

[4] Wang et. al. Interpret neural networks by extracting critical subnetworks. IEEE Transactions on Image Processing 29 (2020), 6707–6720.

**Questions:**

- While the work investigates Faithfulness and Conciseness of the obtained concepts, I would like to ask whether it could be possible whether to know the obtained concepts are sufficient to explain the model? I would suggesting investigating the completeness of the obtained concepts for the global explanation of the model. To do so, the protocol introduced by ShAP could be of interest to use.

- Since the work investigates the extraction of concepts from both CNNs and Vision Transformer, my question is that whether there is any similarity or dissimilarity between obtained concepts from these two different architectures?

- Regarding the issues raised above about the evaluation protocol, it needs to clarify how different methods have been evaluated. Moreover, it is mentioned that the evaluation considers percentage of pixels in the visual parts. Then, I would like to ask how visual parts are selected? since different clusters have different segments related to different images.

---

### Official Review · Reviewer_L5EV · 2024-11-02

**Soundness:** 2
**Presentation:** 3
**Contribution:** 2
**Rating:** 3
**Confidence:** 4

**Summary:**

In this paper, the author introduced a new architecture called Conceptualize Any Network (CAN) that can extract concepts given any neural network. Specifically, the method contains two steps, the first step is to discover patch-based concepts given multiple labeled images. The second step is when given a new unlabelled image, find important concepts for this new image. Moreover, the author did extensive experiments including SDC, SSC, etc. to validate the effectiveness of the proposed method.

**Strengths:**

1. The paper is well-written and easy to understand.
2. The paper provides extensive experiments to show the effectiveness of their proposed method.

**Weaknesses:**

1. In Figure 3, the author compares their result with ACE[1], however, the original paper also shows their result of SDC and SSC that have totally different performances from the author show (60% Acc with only 5 concepts in SSC and 20% Acc with 5 concepts in SDC). Why there is such a huge gap between the original paper and the re-implement results?

2. The method to extract concepts in this paper depends on the other XAI method (Grad-CAM) which limits the novelty and contribution of this paper. It is difficult to measure whether the good performance is the proposed methods or the Grad-CAM.

3. In Table 2, the author compares the conciseness of explanation. However, it is not a fair comparison. As the author said, the MCD has a very big concept hence it has a low conciseness. This means small concepts will cause big conciseness. The author should consider the size of every concept when comparing the conciseness.

[1] Ghorbani, Amirata, et al. "Towards automatic concept-based explanations." Advances in neural information processing systems 32 (2019).

**Questions:**

N/A

---

### Official Review · Reviewer_9eug · 2024-11-03

**Soundness:** 3
**Presentation:** 2
**Contribution:** 2
**Rating:** 5
**Confidence:** 4

**Summary:**

The authors propose a model-agnostic explanation method called Conceptualize Any Network (CAN). The explanation process is divided into two stages: concept discovery, which involves obtaining concepts, and concept assignment, which explains the model's predictions. The authors claim that their method outperforms existing approaches in terms of conciseness and interpretability accuracy.

**Strengths:**

The authors utilize various evaluation criteria to assess both their proposed method and existing methods, which adds robustness to their analysis.

**Weaknesses:**

1) There is a lack of persuasive reasoning behind the design of the proposed methodology. For instance, the rationale for using attribution maps and the design choices for local importance scores are inadequately explained.

2) Questions arise regarding the validity of the authors' claims. Are existing methods truly difficult to apply to models of various architectures?

3) The scope of experimental validation is limited.

The authors assert that their method can generate attribution maps for any architecture, yet only ResNet-50 and ViT_c are experimentally demonstrated. To substantiate their claims, a broader range of models needs to be tested. Additionally, comparisons with other methods for the ViT architecture are entirely absent.

The authors only consider Grad-CAM and Bcos as attribution methods. Various techniques exist for computing attribution maps based on different model architectures, but the paper provides a narrow discussion by only addressing Grad-CAM and Bcos for CNNs and only Bcos for ViTs, failing to explore the implications of different attribution methods comprehensively.

4) There is insufficient persuasion regarding what specifically makes the proposed method superior.

5) The paper presents an explanation method without including examples of the explanations generated, which would enhance clarity and understanding.

**Questions:**

1) To clarify, does the model used to obtain feature attribution correspond to the target model intended for interpretation?

2) It appears that the Bcos method employs a different ResNet model than those provided by the torchvision library. Therefore, is it appropriate to compare CAN using Grad-CAM, MCD, and ACE with CAN using Bcos on the same footing?

3) Why were comparisons with other methods omitted for the ImageNet dataset (as noted in Figure 4)?

4) A broader application of various attribution maps is necessary. Particularly for ViT, where only Bcos is presented, there is a need to apply various attribution maps leveraging attention mechanisms.

---

### Note · Authors · 2024-11-29

I have read and agree with the venue's withdrawal policy on behalf of myself and my co-authors.